# Risks Related to the Use of Non-Steroidal Anti-Inflammatory Drugs in Community-Acquired Pneumonia in Adult and Pediatric Patients

**DOI:** 10.3390/jcm8060786

**Published:** 2019-06-03

**Authors:** Guillaume Voiriot, Quentin Philippot, Alexandre Elabbadi, Carole Elbim, Martin Chalumeau, Muriel Fartoukh

**Affiliations:** 1Assistance Publique-Hôpitaux de Paris, Service de Réanimation médico-chirurgicale, Hôpital Tenon, Hôpitaux Universitaires de l’Est Parisien, 75020 Paris, France; philippot.quentin@gmail.com (Q.P.); alexandre.elabbadi@gmail.com (A.E.); muriel.fartoukh@aphp.fr (M.F.); 2Sorbonne Université, UFR Médecine, 75006 Paris, France; 3Faculté de Médecine, GRC CARMAS, Université Paris Est, 94000 Créteil, France; 4Sorbonne Université, INSERM, Centre de Recherche Saint-Antoine, Team “Immune System, Neuroinflammation and Neurodegenerative Diseases”, Hôpital Saint-Antoine, 75012 Paris, France; carole.elbim@upmc.fr; 5UMR 1153 Epidemiology and Biostatistics Sorbonne Paris Cité Center (CRESS), Obstetrical, Perinatal and Pediatric Epidemiology Research Team (EPOPé), Paris Descartes University, 75014 Paris, France; martin.chalumeau@gmail.com; 6Department of General Pediatrics and Pediatric Infectious Diseases, Necker hospital for Sick Children, Assistance Publique-Hôpitaux de Paris, Université Paris Descartes, 75015 Paris, France

**Keywords:** community-acquired pneumonia, non-steroidal anti-inflammatory drugs, pleural effusion

## Abstract

Non-steroidal anti-inflammatory drugs (NSAIDs) are commonly used to alleviate symptoms during community-acquired pneumonia (CAP), while neither clinical data nor guidelines encourage this use. Experimental data suggest that NSAIDs impair neutrophil intrinsic functions, their recruitment to the inflammatory site, and the resolution of inflammatory processes after acute pulmonary bacterial challenge. During CAP, numerous observational data collected in hospitalized children, hospitalized adults, and adults admitted to intensive care units (ICUs) support a strong association between pre-hospital NSAID exposure and a delayed hospital referral, a delayed administration of antibiotic therapy, and the occurrence of pleuropulmonary complications, even in the only study that has accounted for a protopathic bias. Other endpoints have been described including a longer duration of antibiotic therapy and a greater hospital length of stay. In all adult series, patients exposed to NSAIDs were younger and had fewer comorbidities. The mechanisms by which NSAID use would entail a complicated course in pneumonia still remain uncertain. The temporal hypothesis and the immunological hypothesis are the two main emerging hypotheses. Current data strongly support an association between NSAID intake during the outpatient treatment of CAP and a complicated course. This should encourage experts and scientific societies to strongly advise against the use of NSAIDs in the management of lower respiratory tract infections.

## 1. Non-Steroidal Anti-Inflammatory Drugs (NSAIDs) Are Commonly Used to Alleviate Symptoms during Community-Acquired Pneumonia

Community-acquired pneumonia (CAP) is a frequent disease, with an annual incidence rate ranging from 1.6 to 9 cases per 1000 inhabitants in Western Europe [1,2]. Most patients are treated exclusively in outpatient settings by general practitioners [3]. The mainstay treatment is the early initiation of appropriate antibiotic therapy. However, other medications are widely prescribed for the relief of symptoms such as pain, fever, and cough. Among them, non-steroidal anti-inflammatory drugs (NSAIDs) are commonly used as antipyretics and/or analgesics. In 2001, a prospective survey described the management of community-acquired lower respiratory tract infections by general practitioners in France. NSAIDs were prescribed in half of the pneumonia patients [4]. In another observational study in ambulatory care, half of the patients diagnosed with CAP (*n* = 496) were treated with NSAIDs [5,6]. It is important to note that this frequent use of NSAIDs may have been underestimated because NSAIDs can be obtained over-the-counter in France, as in many countries [7,8]. This extensive use of NSAIDs during the pre-hospital care of CAP patients is supported neither by the guidelines for the management of adult lower respiratory tract infections [9,10,11] nor by clinical data in pneumonia patients [12,13].

## 2. Potential Benefits of NSAIDs during Pneumonia Have Been Explored for Decades, with Conflicting Results

The local inflammatory process involved in microbial clearance during pneumonia is also responsible for parenchymal injury. The polymorphonuclear neutrophils (PMNs), recruited and activated at the site of infection, release reactive oxygen species (ROS), proteolytic enzymes, and anti-microbial peptides. Along with the induced bacterial toxicity, they may damage all the constituents of the alveolar-capillary unit [14]. This inflammatory response involves cyclooxygenases (COX). COX use arachidonic acid to generate prostaglandin H2, which is the precursor of thromboxane A2, prostacyclin, and other prostaglandins. These latter lipid mediators are involved in the recruitment and activation of effector cells such as PMNs [15]. They are two major COX isoenzymes. COX-1 is expressed constitutively in most tissues and is involved in the basal synthesis of the aforementioned lipid mediators, whereas COX-2 is induced during the inflammatory response.

NSAIDs are drugs that block the enzymatic activity of COX. The chemical structure of NSAIDs varies from one to another as does their ability to inhibit each isoenzyme. Therefore, each NSAID is characterized by a COX-selectivity index, which quantifies their selectivity for the COX-2 versus the COX-1 isoenzyme. Aspirin is the least COX-selective drug, whereas celecoxib is one of the most COX-selective NSAIDs.

The inhibitory effects of NSAIDs on PMN functions have been widely described. In vitro, NSAIDs alter adherence, degranulation, phagocytosis, and ROS production by PMNs exposed to various stimuli. In vivo, NSAIDs inhibit the recruitment of PMNs to the inflammatory site and also alter their intrinsic functions [16,17]. In rat models of acute non-specific pleural effusion, pre-treatments with ibuprofen, indomethacin, and flurbiprofen markedly decreased exudate volume and leukocyte migration into pleura [18].

Aside from these COX-dependent mechanisms, NSAIDs display COX-independent effects. Ibuprofen has been shown to inhibit TNFα-induced NFκB transcriptional activity [19]. This may contribute to limiting the local release of pro-inflammatory cytokines such as IL-8, a major chemoattractant of PMNs in humans [20].

Various studies have been conducted to investigate the potential benefits of anti-inflammatory medications in the treatment of pneumonia as the local inflammatory response plays a major role in its pathogenesis. Hence, several NSAIDs have been studied in animal models of pulmonary bacterial infection. Results were conflicting, depending on the drug used, the timing of the administration, the bacterial challenge, and the animal species. In murine models of *Pseudomonas aeruginosa* pulmonary infection, ibuprofen or piroxicam treatment decreased PMN migration and recruitment into the lung as well as the extent of lung tissue damage, and was associated with a lower mortality [21,22,23]. Conversely, acetylsalicylic acid administration before or immediately after the tracheal instillation of *Streptococcus pneumoniae* in mice was associated with both a lower bacterial clearance and a higher mortality [24]. Other studies using NSAIDs in various models of endotoxin-induced extra-pulmonary sepsis also provided conflicting results [25,26,27].

Potential benefits of NSAIDs in relation to gas exchange in pneumonia have also been investigated. Indeed, some prostaglandins display vasodilator effects that may alter hypoxic pulmonary vasoconstriction. This physiological process aims to divert the blood away from a poorly ventilated area such as a lung consolidation. Preliminary studies have suggested that NSAIDs, by restoring the pulmonary hypoxic vasoconstriction, may improve blood oxygenation during pneumonia in dogs [28,29] and humans [30,31]. However, a randomized controlled trial exploring the effects of ibuprofen on gas exchange and airway mechanics during severe sepsis in adults—most of which was pneumonia-related—did not confirm these preliminary results [32].

## 3. NSAIDs Impair the Resolution of Inflammation through COX-2 Inhibition

The pivotal role of COX in the resolution of inflammation, especially inducible COX-2, has been described during the last decade. Briefly, leukotrienes and prostaglandins, generated during the acute phase, stimulate the local release of lipoxins by leukocytes, especially PMNs. These newly formed lipid mediators may interact in an autocrine manner with specific receptors on leukocytes, thereby inhibiting PMN-mediated inflammation and enhancing the phagocytosis of apoptotic PMNs by macrophages. It has been demonstrated that first-phase COX-2-induced eicosanoids promote a shift to anti-inflammatory lipids, which deliver a stop signal [33,34]. This lipid mediator class switching highlights the dual role of COX-2 in the inflammatory response: amplifying the initial acute phase, and then acting for resolution. This dual role has been described in animal studies. In a murine model of acid-induced acute lung injury, selective pharmacologic inhibition or gene disruption of COX-2 was associated with a decreased recruitment of PMNs within the lungs during the acute stage, but was also associated with prolonged lung infiltration and a delayed recovery [35]. In a carrageenan-induced model of pleurisy in rats, both the selective inhibition of COX-2 and the dual inhibition of COX-1/COX-2 limited exudate volume and inflammatory cell recruitment within pleura at 2 h, but exacerbated pleural inflammation at 48 h [36].

## 4. NSAID Exposure during Extra-Pulmonary Infections: A Warning Signal

During bacterial skin and soft tissue infections in adults, observational studies have suggested that the topical or systemic use of NSAIDs is associated with a complicated course [37,38,39,40]. Another report described a strong association between the use of NSAIDs and skin and soft tissue complications (mostly cellulitis and abscess) of varicella zoster virus infection, mostly in children with varicella [41]. Moreover, one recent case-control study identified systemic anti-inflammatory agent exposure as being independently associated with peritonsillar abscess in sore throat [42]. Finally, two additional publications have suggested a role of NSAIDs in promoting complications such as the dissemination of infection to more than one site or suppuration, and delaying the prescription of effective antibiotic therapy during miscellaneous bacterial infections requiring hospitalization [43,44].

## 5. NSAID Exposure during Pneumonia: Numerous Studies Support a Risk of Complicated Course

As reported above, the two theoretical means of interest in using NSAIDs during lower respiratory tract infections provided conflicting results in both experimental and human studies. Hence, the scientific interest in NSAIDs for the treatment of pneumonia has decreased over time. At the same time, clinical data have been reported that support a statistical association between NSAID exposure and a complicated course of CAP requiring hospitalization (Table 1).

### 5.1. In Hospitalized Children

In the United States, a retrospective study reported an increasing incidence of pleural empyema between 1993 and 1999, rising from one to five cases per 100,000 people aged less than 19 years old. Among children hospitalized for CAP during the study period, 28% had pleural empyema. NSAID exposure before hospitalization was identified as an independent risk factor of pleural empyema [46]. Consistently, a retrospective study conducted in two French hospitals reported an increasing incidence of complicated pneumonia, defined as a pleural effusion and/or a lung cavitation in children. Between 1995 and 1999, 3% of hospitalized cases of pneumonia were complicated when compared with 23% in 2003. Multivariate analysis identified ibuprofen exposure as the only pre-hospital treatment independently associated with a higher risk of complicated pneumonia [45]. In a British prospective cohort of 160 children hospitalized for CAP between 2009 and 2011, 40 developed a pleural empyema. Pre-hospital NSAID exposure was as high as 82% in these children who developed pleural empyema, compared with 46% in uncomplicated cases [49]. More recently, a prospective monocentric study conducted in Poland included 203 children hospitalized for CAP between 2012 and 2014. Forty-two developed a pleural or pulmonary complication including para-pneumonic pleural effusion, pleural empyema, necrotizing pneumonia, and lung abscess. Ibuprofen exposure was identified as an independent risk factor of complication. A dose–effect relationship was described: exposure to a cumulative dose of ibuprofen higher than 78 mg/kg was associated with an increased risk of pleural or pulmonary complication [53].

All these pediatric studies shared the methodological weakness of not accounting for protopathic bias [55]. This bias occurs in case-control epidemiological studies when it is difficult to determine if the exposure to the studied factor preceded the occurrence of the complication. Hence, in these case-control studies, NSAIDs could have been prescribed because of symptoms (thoracic pain, fever) related to the beginning of a complication (parapneumonic pleural effusion, pleural empyema). Therefore, NSAID exposure may only be a marker, rather than a cause of the occurrence of a pleural or a pulmonary complication. In order to account for this bias, a case-control study was conducted in 15 French centers between 2006 and 2009. The cases involved consecutive children hospitalized for a pleural empyema occurring in the 15 days following a viral-associated respiratory tract infection treated at home. The controls were children with a viral-associated respiratory tract infection treated at home who did not require hospitalization. Controls were matched for the general practitioner (same physician), age, viral symptoms, and season. Eighty-three case-control pairs were studied. Infection was localized in the lower respiratory tract in 23% of cases and 34% of controls (*p* = 0.21). NSAID exposure, almost exclusively to ibuprofen, was involved in 39% of cases and 27% of controls (*p* = 0.08). Half of the children received acetaminophen (47% vs. 49%; *p* = 0.79). Eight percent of the cases and 15% of the controls received antibiotics concomitantly to the first day of viral symptoms (*p* = 0.21). In multivariate analysis, an NSAID treatment starting in the first three days of viral symptoms and administered for at least one day was independently associated with a higher risk of pleural empyema (OR, 2.79; 95% confidence interval (CI), 1.40–5.58). An antibiotic therapy started within the first three days of viral symptoms and administered for at least six days was independently associated with a lower risk of pleural empyema (OR, 0.32; 95% CI, 0.11–0.97). In the sub-group of children exposed to NSAIDs, the risk of pleural empyema was increased if the duration of antibiotic therapy was less than six days (OR, 3.01; 95% CI, 1.52–5.95) [50].

### 5.2. In Adults Admitted to Intensive Care Units (ICUs)

In 2011, our group reported a prospective cohort of 90 consecutive patients admitted to ICUs for CAP. More than one third of them had received a pre-hospital treatment with NSAIDs. The duration of NSAID exposure was 5 ± 2 days. The patients exposed to NSAIDs were younger (47 vs. 55 years; *p* = 0.017) and had fewer comorbidities (6% vs. 30%; *p* = 0.003). NSAID exposure was associated with a longer duration of symptoms before hospital referral (4.7 ± 2.5 days vs. 3.5 ± 2.5 days; *p* = 0.03) and ICU referral (5.5 ± 2.4 days vs. 4.1 ± 2.5 days; *p* = 0.009). At hospital admission, exposed patients had more clinical and radiological signs of pleural effusion (41% vs. 14%; *p* = 0.009 and 53% vs. 22%; *p* = 0.006, respectively). During ICU stay, exposed patients more often developed pleuropulmonary complications such as pleural empyema and lung cavitation (38% vs. 5%; *p* = 0.0009) and showed a tendency toward more invasive diseases, with a higher frequency of pleural empyema (25% vs. 5%; *p* = 0.014) and bacteremia (34% vs. 28%; *p* = 0.56), especially those who had not received concomitant antibiotic therapy (69% vs. 27%; *p* = 0.009). In multivariate analysis, NSAID exposure was identified as an independent factor of pleuropulmonary complications (OR, 8.1; 95% CI, 2.3–28). Finally, the mean duration of antimicrobial therapy administered for pneumonia was 10 days longer in exposed patients (24.5 days vs. 15.4 days, *p* = 0.003), consistent with a trend toward longer ICU and hospital lengths of stay [47]. Three years later, another French group reported a retrospective cohort of 106 adult patients admitted to ICUs for pneumococcal CAP. Twenty patients were exposed to NSAIDs prior to hospital referral including 17 (85%) on medical prescriptions. As described before, exposed patients were younger and more active, and had less comorbidities. NSAID exposure was associated with a delayed hospital referral and a higher frequency of pleural effusion [48].

### 5.3. In Hospitalized Adults

In 2017, Basille et al. reported a prospective cohort of 221 patients hospitalized for CAP in France. Twenty-four (11%) were exposed to NSAIDs prior to hospital referral. Exposed patients were younger (50 vs. 67 years old, *p* = 0.001) and had fewer comorbidities (Charlson score = 0: 60% vs. 25%; *p* = 0.001). The time until the administration of an effective antibiotic therapy, defined as the time from the first symptoms of pneumonia to the first dose of appropriate antibiotics, was 3.3 days longer in exposed patients. In multivariate analysis, pre-hospital NSAID exposure was independently associated with complicated pneumonia (OR 2.57; 95% CI, 1.02–6.64), defined as the occurrence of lung abscess or complicated parapneumonic pleural effusion [52]. One year later, Basille et al. conducted a registry-based study in Denmark. They identified 59,250 patients admitted to hospital with a first-time diagnosis of CAP in the period 1997–2011. The NSAID users (*n* = 9012) were split into new NSAID users (patients with a first-ever NSAID prescription within 60 days before hospital admission, *n* = 2294) and long-term NSAID users (patients who had filled a previous NSAID prescription 61–365 days before hospital admission, *n* = 6718). Other patients included former users and nonusers. New NSAID users were younger and had fewer comorbid conditions when compared with long-term users and former users. NSAID users had a higher risk of pleuropulmonary complications (pleural empyema or lung abscess) (3.8%) in comparison with both former users (2.4%) and non-users (2.3%) (adjusted rate ratio (aRR) 1.81; 95% CI, 1.60–2.05). Among the NSAID users, a stratified analysis showed the highest adjusted RR of pleuropulmonary complications in young patients (aRR 3.48; 95% CI, 2.64–4.60) and in patients without comorbidities (aRR 2.29; 95% CI, 1.94–2.70). The statistical association between NSAID exposure and the occurrence of pleuropulmonary complications was also described in the subgroup of long-term NSAID users and in new NSAID users, after the exclusion of all patients who had collected their last prescription of NSAIDs within 10 days before hospital admission. Based on these findings, authors assumed that the statistical association they observed was not solely explained by protopathic bias [54]. Finally, a single center prospective study of 57 patients with a diagnosis of pneumonia and parapneumonic pleural effusion was conducted in Greece. Multivariate analysis identified a pre-hospital NSAID exposure of more than six days as an independent risk factor of prolonged hospitalization [51].

## 6. Synthesis of Human Data and Mechanistic Hypotheses

During CAP, numerous observational data collected in hospitalized children [45,46,49,50,53], hospitalized adults [51,52,54], and ICU adults [47,48] supported a strong association between pre-hospital NSAID exposure and (i) a delayed hospital referral, (ii) a delayed administration of antibiotic therapy, and (iii) the occurrence of pleuropulmonary complications, even in the only study that accounted for a protopathic bias [50]. Other endpoints have been described including a longer duration of antibiotic therapy [47] and greater hospital length of stay [47,51]. In all adult series, patients exposed to NSAIDs were younger and had fewer comorbidities [47,48,52,54]. These data suggest a higher use of NSAIDs in healthy people, but without deciphering the respective roles of self-medication and medical prescription.

The mechanisms by which NSAID use would entail a complicated course in pneumonia still remain uncertain. Based on the observational data exposed above, two main hypotheses have emerged:
-Temporal hypothesis: By alleviating the major symptoms of inflammation such as fever and pain, NSAID intake might impede the timely recognition and delay the diagnosis of pneumonia and the subsequent initiation of an appropriate antibiotic therapy. The delayed treatment may promote a more invasive disease, with a higher frequency of pleural empyema and bacteremia.-Immunological hypothesis: During pneumonia, NSAIDs may limit the local recruitment of innate immune cells and alter the intrinsic functions of PMNs including phagocytosis and ROS production (Figure 1). This would result in a lower bacterial clearance and a local noncontainment of the infectious process. This alteration of the immune response may promote multilobar and/or bilateral pneumonia as well as cavitation and pleural effusion. Moreover, the inhibition of COX-2-induced lipid mediator class switching may extend the acute phase and delay the resolution of inflammation (Figure 1). This would result in a prolonged disease with delayed clinical stability and a longer hospital length of stay.

Future studies are needed to address the causal role of NSAIDs in promoting a complicated course in pneumonia. Translational research in CAP inpatients might provide answers. Quantitative and qualitative analysis of alveolar fluid samples might display a blunted pulmonary recruitment of PMNs, an alteration of their cellular functions, and a lower bacterial clearance in NSAID-exposed patients. Blood analysis might show a limited release of inflammation mediators during the initial phase but a sustained inflammation during the later phase, with delayed lipid mediator class switching. However, such studies are difficult to perform and their results difficult to interpret. Access to human samples, especially from the lungs, is limited. Biological studies on human PMNs are technically challenging and require fresh samples with which to work. Finally, NSAID-exposed patients are heterogeneous in terms of dose regimen, treatment duration, and delay to hospital referral.

## 7. Conclusions

NSAIDs are frequently used as a symptomatic treatment during lower respiratory tract infections (LRTIs) in adults and children, while neither clinical data nor guidelines encourage this use. Numerous observational series of CAP inpatients support a strong association between a pre-hospital NSAID exposure and a protracted and complicated course of pneumonia. These data should encourage experts and scientific societies to strongly advise against the use of NSAIDs in the management of LRTIs.

## Figures and Tables

**Figure 1 jcm-08-00786-f001:**
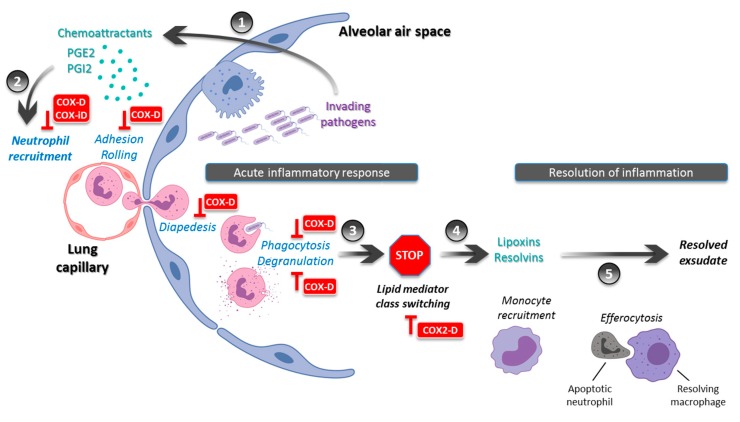
During pneumonia, NSAIDs may interfere with acute inflammation response and resolution. Invading pathogens within the alveolar air space induce an innate immune response, which involves cyclooxygenases (COX) to generate lipid mediators of inflammation such as prostaglandins (PG) E2 and I2 (**1**). NSAIDs may alter the subsequent recruitment of polymorphonuclear neutrophils through both COX-dependent and COX-independent effects. NSAIDs also alter their intrinsic functions such as phagocytosis and degranulation (**2**). The acute phase response is followed by a lipid mediator class switching (**3**), which leads to the biosynthesis of pro-resolving mediators such as lipoxins and resolvins (**4**). By inhibiting this stop signal through COX-2 inhibition, NSAIDs may limit the local recruitment of monocytes, which act through efferocytosis to resolve exudate, and subsequently hamper the resolution of inflammation (**5**). Abbreviations: COX-D = Cyclooxygenase-dependent; COX-iD = Cyclooxygenase-independent; COX2-D = Cyclooxygenase 2-dependent. Adapted from Serhan et al. [34].

**Table 1 jcm-08-00786-t001:** Studies exploring the impact of pre-hospital exposure to NSAIDs in lower respiratory tract infections.

Author; Study Period	Population (Number of Patients); Country	Study Drug	Care Setting	Study Design	Comment
Francois [45] 1995–2003	Children (*n* = 767), CAP, France	Ibuprofen	Hospital ward	Retrospective case-control	Recent NSAID exposure was an independent risk factor of pleural empyema (OR 2.6 (1.5–4.4))
Byington [46] 1993–1999	Children (*n* = 540), CAP, United States	Ibuprofen	Hospital ward	Retrospective case-control	Recent NSAID exposure was an independent risk factor of pleural empyema (OR 4.0 (2.5–6.5))
Voiriot [47] 2002–2006	Adults (*n* = 90), CAP, France	NSAIDs	ICU	Prospective cohort	Recent NSAID exposure was an independent risk factor of pleuropulmonary complications (pleural empyema, excavation) (OR 8.1 (2.3–28))
Messika [48] 1997–2009	Adults (*n* = 106), pneumococcal CAP, France	NSAIDs	ICU	Historical cohort	Recent NSAID exposure was associated with a higher risk of pleuropulmonary complications (pleural empyema, excavation) (OR 5.8 (2.0–17))
Elemraid [49] 2009–2011	Children (*n* = 160), CAP, UK	Ibuprofen	Hospital ward	Prospective case-control	Recent NSAID exposure was involved in 82% of cases with pleural empyema, compared to 46% of cases without complications (OR 1.9 (0.8–3.2))
Le Bourgeois [50] * 2006–2009	Children (*n* = 83), viral LRTI, France	NSAIDs	Hospital ward	Prospective case-control	Recent NSAID exposure was an independent risk factor of pleural empyema (OR 2.8 (1.4–5.6))
Kotsiou [51] 2015–2016	Adults (*n* = 57), CAP, Greece	NSAIDs	Hospital ward	Prospective cohort	Pre-hospital NSAID use for more than 6 days was associated with a prolonged hospitalization duration
Basille [52] 2008–2013	Adults (*n* = 221), CAP, France	NSAIDs	Hospital ward	Prospective cohort	Recent NSAID exposure was an independent risk factor of pleural empyema (OR 2.6 (1.02–6.6))
Krenke [53] 2012–2014	Children (*n* = 203), CAP, Poland	Ibuprofen	Hospital ward	Prospective cohort	A dose–effect relationship was found: exposure to a cumulative dose of ibuprofen higher than 78 mg/kg was significantly associated with an increased risk of pleuropulmonary complications, such as parapneumonic pleural effusion, pleural empyema, necrotizing pneumonia and pulmonary abscess (OR 2.5 (1.3–4.9))
Basille [54] * 1997–2011	Adults (*n* = 59,250), CAP, Denmark	NSAIDs	Hospital ward	Registry-based	NSAID exposure was associated with pleural empyema and/or lung abscess (RR 1.81 (1.60–2.05))

Abbreviations: CAP = community acquired pneumonia; ICU = intensive care unit; LRTI = lower respiratory tract infections; *n* = number; NSAIDs = non-steroidal anti-inflammatory drugs; OR = odds ratio; RR = rate ratio. Odds ratio (OR) and rate ratio (RR) are expressed as values at 95% confidence intervals. * Studies that took into account the protopathic bias.

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
