# Peer review of "Risks Related to the Use of Non-Steroidal Anti-Inflammatory Drugs in Community-Acquired Pneumonia in Adult and Pediatric Patients"

_jcm, 2019, doi:10.3390/jcm8060786_

Round 1
Reviewer 1 Report
It is an interesting review. There are not many about these drugs and the risk of pneumonia.
I have 2 comments:1) I would like more details about pathogenesis of the interaction NSAID -pneumonia.
2)what about the interaction of statins and NSAID - pneumonia?
3)I believe we need a paragraph about the future studies needed and why it is difficult..
Author Response
Reviewer 1 Round 1
It is an interesting review. There are not many about these drugs and the risk of pneumonia.
1) I would like more details about pathogenesis of the interaction NSAID-pneumonia.
To address this comment, we have built a new figure (Figure 1), which illustrates the interactions NSAIDs-pneumonia. Title and legend of this figure have been added to the end of the manuscript.
2) What about the interaction of statins and NSAID- pneumonia?
This is a very interesting point. Unfortunately, as far as you know, there’s no data illustrating a putative role of statin in preventing (or promoting/worsening) NSAIDs-associated pleuropulmonary complications in CAP.
3) I believe we need a paragraph about the future studies needed and why it is difficult.
We are grateful for this comment. A paragraph has been added in the manuscript in the Section 6, as followed:
“Future studies are needed to address the causal role of NSAIDs in promoting a complicated course in pneumonia. Translational research in CAP inpatients might give answers. Quantitative and qualitative analysis of alveolar fluid samples might display a blunted pulmonary recruitment of PMNs, an alteration of their cellular functions, and a lower bacterial clearance in NSAIDs-exposed patients. Blood analysis might show a limited release of mediators of inflammation during the initial phase but a sustained inflammation during the late phase, with a delayed lipid mediator class switching. However, such studies are difficult to perform and their results difficult to interpret. Access to human samples, especially from the lungs, is limited. Biological studies on human PMNs are technically challenging and suppose to work on fresh samples. Finally, NSAIDs-exposed patients are heterogeneous in terms of dose regimen, treatment duration and delay to hospital referral.”
Reviewer 2 Report
-- Very interesting topic and has potential to deliver a strong message but the paper will need extensive editing before it is in a position to do that
-- Would recommend revising the introduction and conclusion closely to make the paper more interesting, appealing and to deliver a strong message
-- Paper will require extensive editing. Both for English style and grammar
-- The only table used in the paper is not well organized and has errors
Please find the detailed comments in the attached file.

Author Response
Reviewer 2 Round 1
- Very interesting topic and has potential to deliver a strong message but the paper will need extensive editing before it is in a position to do that.
- Would recommend revising the introduction and conclusion closely to make the paper more interesting, appealing and to deliver a strong message.
- Paper will require extensive editing. Both for English style and grammar.
- The only table used in the paper is not well organized and has errors.
Many thanks for this careful reviewing, which allowed to improve the manuscript. Please find below a point-by-point response.
1) Recommend re-structuring the whole paragraph. The authors are trying to make the point that NSAIDs are frequently used in the management of CAP, however, the point is not being made properly.
Both first and second paragraph need re-structuring.
As suggested, the first part of the manuscript has been rewritten and shortened, in order to focus on the point (NSAIDs are commonly used to alleviate symptoms during CAP):
“Community-acquired pneumonia (CAP) is a frequent disease, with an annual incidence rate ranging from 1.6 to 9 cases for 1000 inhabitant in western Europe (1,2). Most patients are treated exclusively in outpatient settings by general practitioners (3). The mainstay treatment is the early initiation of appropriate antibiotic therapy. However, other medications are widely prescribed for relief of symptoms such as pain, fever and cough. Among them, non-steroidal anti-inflammatory drugs (NSAIDs) are commonly used as antipyretics and/or analgesics. In 2001, a prospective survey described the management of community-acquired lower respiratory tract infections by general practitioners in France. The prescriptions of analgesics, antipyretics and antitussives accounted for two-third of all prescriptions. Most of all, NSAIDs were prescribed in half of the pneumonia patients (4). In another observational study in ambulatory care, half of the patients diagnosed with a CAP (n=496) were treated with NSAIDs (5,6). Of note, this frequent use of NSAIDs may have been underestimated, because NSAIDs can be obtained over-the-counter in France as in many countries (7,8). This large use of NSAIDs during pre-hospital care of CAP patients is not supported neither by guidelines for the management of adult lower respiratory tract infections (9–11) nor by clinical data in pneumonia patients (12,13).”
2) Line 58: “for decades”
This has been corrected.
3) Line 85-86: Would recommend changing the sentence. Can try: "Various studies have been conducted to investigate the potential benefit/role of anti-inflammatory medications in treatment of pneumonia as local inflammatory response plays a major role in its pathogenesis."
The sentence has been corrected as suggested.
4) Line 110-112: Would make a point about how NSAIDS and inhibiting the COX-2 would impair the resolution of inflammation!
To address this comment (and comment 1, Reviewer 1), we have built a new figure (Figure 1), which illustrates how NSAIDs (and inhibition of COX-2) would impair the resolution of inflammation during pneumonia. Title and legend of this figure have been added to the end of the manuscript.
5) Line 130: Not sure what this means or implies
To address this comment, titles of paragraphs 4 and 5 have been changed as followed:
4. NSAIDs exposure during extra-pulmonary infections: a warning signal
5. NSAIDs exposure during pneumonia: numerous studies support a risk of complicated course
6) Line 131: Would avoid using the word “exposed”
The word “exposed” has been changed for “reported”.
7) Table 1: Not sure if authors are trying to say Odds ratio was 2.6 with Confidence Interval ranging between 1.5-4.4
We are grateful for this comment. Results in the right column of the table are expressed as Rate ratio/Odds ratio [95% confidence interval]. We added a mention in the legend of the table, as followed: “Odds ratio (OR) and rate ratio (RR) are expressed as value [95% confidence interval]”. Moreover, to improve its presentation in a portrait format, the table has been reorganized: some columns have been merged and the mention of the protopathic bias has been moved to the table legend (“ *Studies having taken account the protopathic bias”).
8) Line 143: per 100,000 cases.
The word “for” has been changed for “per”.
9) Line 148: Too much use of the word “indeed” throughout the paper.
This occurrence of “indeed” has been deleted.
10) Line 148, “hospitalized cases”: Hospitalized cases of what??
”Hospitalized cases” has been changed for “Hospitalized cases of pneumonia”.
11) Line 244-249: Would recommend changing the structure of the paragraph. But it should be noted that it is a very interesting paragraph. However, would this paragraph to be powerful in wording and structure.
As suggested, this paragraph has been restructured and rewritten, and mention of future studies has been added as suggested by Reviewer 1:
“During CAP, numerous observational data collected in hospitalized children (47,48,51,52,55), hospitalized adults (53,54,56) and ICU adults (49,50) support a strong association between pre-hospital NSAIDs exposure and i) a delayed hospital referral, ii) a delayed administration of antibiotic therapy, and iii) the occurrence of pleuropulmonary complications, even in the only study which accounted for a protopathic bias (52). Other endpoints have been described, including longer duration of antibiotic therapy (49) and hospital length of stay (49,53). In all adult series, patients exposed to NSAIDs were younger and had fewer comorbidities (49,50,54,56). These data suggest a higher use of NSAIDs in healthy people, but without deciphering the respective roles of self-medication and medical prescription.
The mechanisms by which the NSAIDs use would entail a complicated course in pneumonia still remain uncertain. Based on the observational data exposed above, two main hypotheses are emerging
- Temporal hypothesis: With alleviating the major symptoms of inflammation such as fever and pain, NSAIDs intake might impede the timely recognition and delay the diagnosis of pneumonia and the subsequent initiation of an appropriate antibiotic therapy. The delayed treatment may promote a more invasive disease, with a higher frequency of pleural empyema and bacteremia.
- Immunological hypothesis: During pneumonia, NSAIDs may limit the local recruitment of innate immune cells and alter the intrinsic functions of PMNs including phagocytosis and ROS production. It would result in a lower bacterial clearance and a local noncontainment of the infectious process. This alteration of the immune response may promote multilobar and/or bilateral pneumonia, as well as cavitation and pleural effusion. Moreover, the inhibition of the COX-2-induced lipid mediators class switching may extend the acute phase and delay the resolution of inflammation. It would result in a prolonged disease, with a delayed clinical stability and a longer hospital length of stay.
Future studies are needed to address the causal role of NSAIDs in promoting a complicated course in pneumonia. Translational research in CAP inpatients might give answers. Quantitative and qualitative analysis of alveolar fluid samples might display a blunted pulmonary recruitment of PMNs, an alteration of their cellular functions, and a lower bacterial clearance in NSAIDs-exposed patients. Blood analysis might show a limited release of mediators of inflammation during the initial phase but a sustained inflammation during the late phase, with a delayed lipid mediator class switching. However, such studies are difficult to perform and their results difficult to interpret. Access to human samples, especially from the lungs, is limited. Biological studies on human PMNs are technically challenging and require to work on fresh samples. Finally, NSAIDs-exposed patients are heterogeneous in terms of dose regimen, treatment duration and delay to hospital referral.
12) Line 255-257: Sentence structure and grammar needs to be corrected
This sentence, which was supporting the main message, has been deleted from the conclusion.
Round 2
Reviewer 2 Report
Significant improvements have been made but will still need some changes in the style of presentation. Recommend further editing of the introduction and conclusion paragraph.
Author Response
To the Editor of Journal of Clinical Medicine:
As recommending by Reviewer 2, we ordered an English editing (via MDPI) of the whole manuscript.
Many thanks
Best regards
